# A Novel Target Detection Method Based on Multi-Parameter Space for Mobile Passive Multistatic Radar

Hua Zhang [1,2], Yiheng Liu [1,2], Qinghai Dong [1], Ning Liu [3], Kai Chang [3], Xuemei Wang [1,*] and Xiaode Lyu [1,2]

1   National Key Laboratory of Microwave Imaging Technology, Aerospace Information Research Institute, Chinese Academy of Sciences, Beijing 100094, China; zhanghua211@mails.ucas.ac.cn (H.Z.); liuyiheng17@mails.ucas.ac.cn (Y.L.); dongqh@aircas.ac.cn (Q.D.); lvxd@aircas.ac.cn (X.L.)
2   School of Electronic, Electrical and Communication Engineering, University of Chinese Academy of Sciences, Beijing 100094, China
3   Northern Institute of Electronic Equipment, Beijing 100083, China; lning1965@126.com (N.L.); kerkaichance@gmail.com (K.C.)
*   Correspondence: wangxm003775@aircas.ac.cn

**Abstract:** Deploying Passive Multistatic Radar (PMR) on mobile platforms provides covert and cost-effective monitoring over a large area, offering certain advantages in countermeasure. However, mobile PMR faces significant challenges, such as Doppler distortion and phase deviations. A multi-parameter space target detection method is proposed for mobile PMR to achieve target detection in three-dimensional environments. By estimating the Doppler Frequency Rate (DFR), applying bistatic range phase compensation, and implementing azimuth time integration, frame division, and data fusion, the detection accuracy and the Signal-to-Noise Ratio (SNR) are improved. Simulation results indicate that the proposed method significantly enhances the SNR and produces accurate detection results, demonstrating its efficacy.

**Keywords:** mobile passive multistatic radars; Doppler frequency rate; multi-parameter space; target detection





## 1. Introduction

Passive radar systems detect targets by utilizing electromagnetic signals emitted from illuminators of opportunity [1], rather than transmitting their own signals. Currently, these systems commonly use static illuminators like broadcast television signals and communication base stations. This technology has reached a relatively mature stage, with various methods such as the linearization method based on Time-Difference-Of-Arrival (TDOA) and Angle-Of-Arrival (AOA) parameters [2–4]. Other notable methods include the quasi-Newtonian spectral method of nonlinear equation algorithms [5], and the utilization of Kalman filter class [6] and its extended class methods [7]. Recent advancements have also opened up new perspectives and innovative research areas for investigation and practical applications [8,9]. One of the most challenging issues in deploying PMR systems on mobile platforms is the estimation of Doppler frequency, which can be affected by Doppler coupling [10], Doppler distortion [11] and Doppler resolution [12,13]. These effects can result in adverse outcomes such as low SNR, increased errors, and the emergence of ghost targets. Another problem arises when multiple illuminators of opportunity are in motion at different locations, as fusing the data from these moving signals may introduce phase errors in the received echoes. Without a common reference for fusing multiple echoes, the ability to detect targets cannot be significantly enhanced.

Due to platform motion, Doppler distortion and coupling occur in the signals. Previous research has extensively explored this issue, as documented in [14–17]. Ummenhofer [14] specifically studied the use of signal characteristics from Digital Video Broadcasting-Terrestrial (DVB-T) and standard Orthogonal Frequency Division Multiplexing (OFDM)

to estimate Doppler effects in situations where the receiver experiences highly nonlinear motion. However, this study did not consider the Doppler distortion caused by the movement of illuminators of opportunity, as they were assumed to be stationary in the scenario. In a different study, Mixon [15] investigated the direct localization problem of Doppler frequency shift based on illuminators of opportunity when both the receiver and transmitters are mobile. The study proposed a direct localization algorithm that converts the maximum likelihood function for target position estimation into a cost function represented by eigenvalues. However, it is important to note that this method is limited to scenarios with a small number of sample points and targets that move slowly in relation to the illuminators of opportunity. Huang [16] investigated the problem of varying range migration and Doppler parameters during echoes in different motion stages. In his study, a conjugate integral processing method was used to accumulate target energy in the Doppler Center (DC) and Doppler frequency rate domain for each stage. By projecting and combining the compensated DC-DFR map in the high-dimensional Doppler parameter space, the issue caused by multi-level Doppler parameters was effectively resolved. Palmer [17] examined the issue of reference signal mismatch on a mobile receiver platform. By reconstructing the reference signal to compensate for the Doppler distortion caused by transmitter mismatch and motion, the combined effects of these two factors were effectively addressed. The main objective of this study was to estimate the Doppler frequency rate by estimating the target bistatic range.

Recent studies have utilized satellite systems as external radiation sources for target detection. Satellites are advantageous in this regard due to their distance from the ground, which makes it easier to approximate the Doppler frequency rate [18–21]. Duan [19] constructed a pre-compensation function using the range and Doppler of the beam center pointing cell, which mitigated the Doppler frequency rate caused by different observation configurations. In a study conducted by Wen [21], the focus was on examining the acquisition of multi-station echoes in the Cartesian Doppler frequency rate domain. The study explored a single transmission scenario and proposed the design of a three-dimensional sliding window. This sliding window was aimed at achieving optimal matching of specified targets in any motion direction. The findings of this study provided a foundation for the subsequent fusion process.

The aforementioned studies have provided valuable insights that have inspired our own study. However, it is important to note that these studies have primarily focused on two-dimensional (2D) planes or static illuminators of opportunity. In order to address the challenges posed by Doppler distortion and data fusion in a three-dimensional (3D) space scenario, our paper aims to investigate the performance of detecting targets using a PMR system with moving multiple illuminators of opportunity. Building upon the method described in [16,18] that utilizes Doppler frequency rate information of echoes, we have developed a mobile PMR system model and expanded the multi-parameter space target detection method to 3D space.

This paper presents two main contributions. Firstly, it proposes a model for a 3D mobile PMR system. In this model, the receiver and the illuminators of opportunity are situated on the same plane and move at low velocities, while the target is located on a distant space and moves at a high velocity. Secondly, the paper derives the relationship between the Doppler frequency rate and the target's equivalent radial velocity and radial acceleration. Additionally, the paper proposes a method to enhance the SNR by fusing the signals from multiple illuminators of opportunity.

The remainder of this paper is organized as follows: Section 2 deduces a mobile PMR system model. Section 3 introduces the proposed target detection method. In Section 4, simulation results are presented to demonstrate the effectiveness of the method. Finally, the discussion and conclusion are provided in Section 5 and 6. The flowchart of the data fusion of target detection with multiple signals is presented in Figure 1.

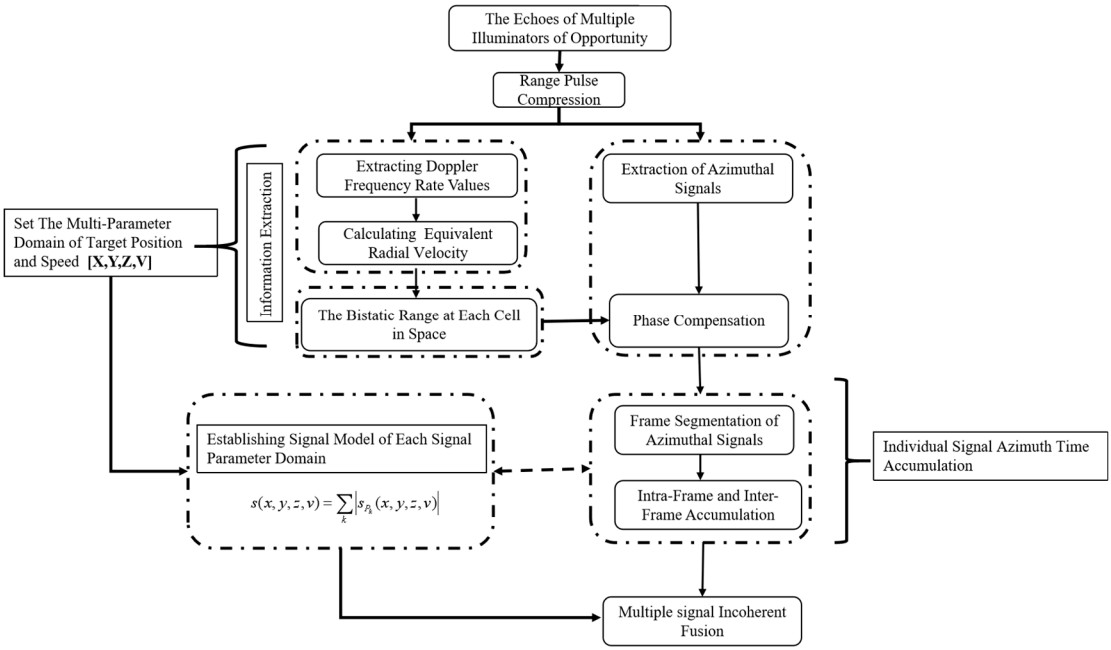

**Figure 1.** The flowchart of target detection with multiple signals fusion.

## 2. Mobile Passive Multistatic Radar System Model

The model proposed for detecting maritime moving targets using the BeiDou satellites as illuminators of opportunity [18] relies on the long distance between the satellite, receiver, and target. However, due to its large baseline angle, errors may arise, rendering it unsuitable for target detection in a short baseline scenario in 3D space. In the model proposed in this paper, the receiver is located closer to the illuminators of opportunity, while the target is farther away, as shown in Figure 2, and a Cartesian coordinate system is established based on the right-hand rule [22]. In this case, the baseline of the model is short, and the Doppler frequency rate needs to be approximated by other methods. The parameters in Figure 2 are described below: $P_k$ denotes the k-th illuminator of opportunity, $O$ denotes the moving receiver, $R_{O,k}$ denotes the baseline between $P_k$ and $O$, and $R_{O,T}$ and $R_{P_k,T}$ denote the distances from $O$ and $P_k$ to the target $T$, respectively. The parameters of the receiver are $(0, 0, 0, v)$, the parameters of the target are $(x_T, y_T, z_T, v_T, a_T)$, and the parameters of the k-th illuminator of opportunity are $(x_{P_k}, y_{P_k}, 0, v_{P_k}, a_{P_k})(k = 1, 2, 3 \ldots)$.

Assuming that the transmitted signals from illuminators of opportunity are continuous signals and independent of each other, as shown in Figure 3,

$$s_{P_k}(t) = A_k \cdot \exp(j2\pi f_c t + \varphi_{0,k})(k = 1, 2, 3) \tag{1}$$

The amplitude of the transmitted signals from different illuminators of opportunity is denoted as $A_k$, k denotes the k-th illuminator of opportunity, $t$ represents the time delay in the range direction, the carrier frequency is denoted as $f_c$, and the corresponding initial phase is denoted as $\varphi_{0,k}$. The reflected echo signals from the target experience a delay and have different amplitudes. This delay is manifested in their cross-correlation function. Cross-correlating the direct signal and the echo signal yields a range pulse compression signal that is denoted as

$$s_{pc,P_k}(t, \eta) = \sigma_{\eta,k} \exp\left\{-j2\pi f_c \frac{R_k(\eta)}{c}\right\} \times r\left[t - \frac{R_k(\eta)}{c}\right] \tag{2}$$

where $\eta$ represents the time delay in the azimuth direction, $\sigma_{\eta,k}$ denotes the complex scattering coefficient of the target, $c$ represents the velocity of light, and $r(\cdot)$ denotes the cross-correlation function between the echo signal and the direct signal. $R_k(\eta)$ represents

the bistatic range, and it is equivalent to $R_k(\eta) = R_{O,T}(\eta) + R_{P_k,T}(\eta) - R_{O,k}(\eta)$. The phase in the range pulse compression signal is denoted as $\phi(\eta) = -f_c \frac{2\pi}{c} R_k(\eta)$, and taking the derivative with respect to the time delay in the azimuth direction yields

$$\frac{1}{2\pi} \frac{d\phi(\eta)}{d\eta} = f_d(\eta) = \overline{f}_{dc,k} + \overline{f}_{dt,k} \cdot \eta\left(|\eta| \leq \frac{T}{2}\right) = -\frac{f_c}{c} \overset{\bullet}{R_k}(\eta) \tag{3}$$

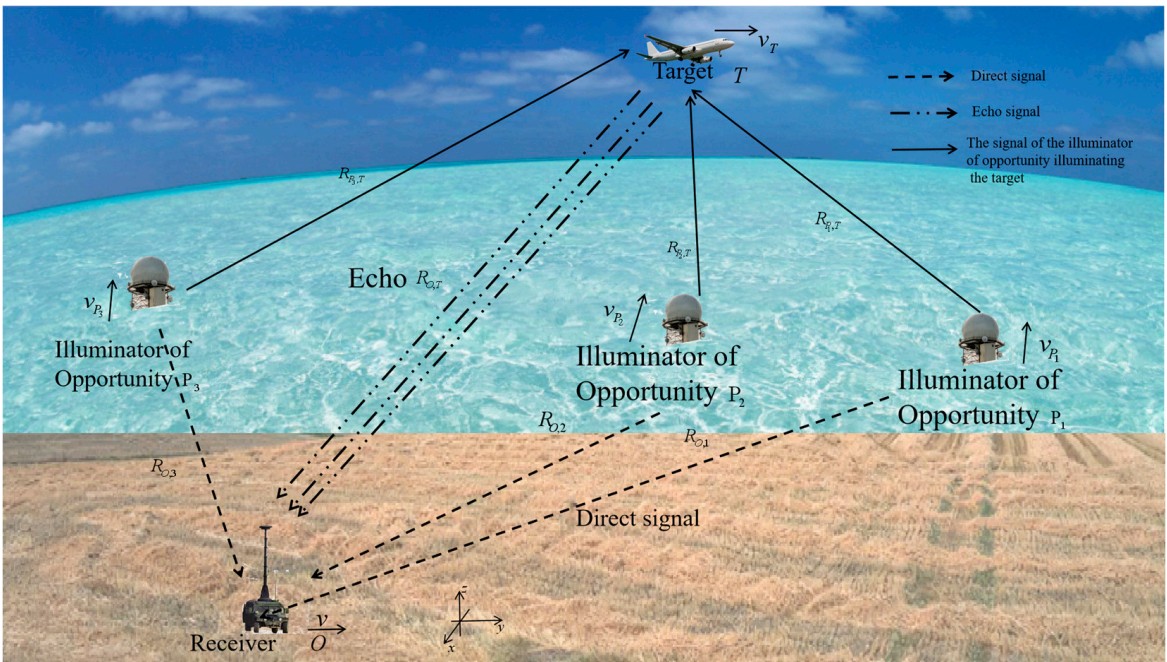

**Figure 2.** Mobile PMR system model with three illuminators of opportunity. This is an example where the illuminators of opportunity are actually close to the receiver, and the baseline is very short. Multiple illuminators of opportunity are in motion on the sea surface, and the receiver has the capability to receive direct signals of different illuminators of opportunity in the same plane and echoes reflected by the target.

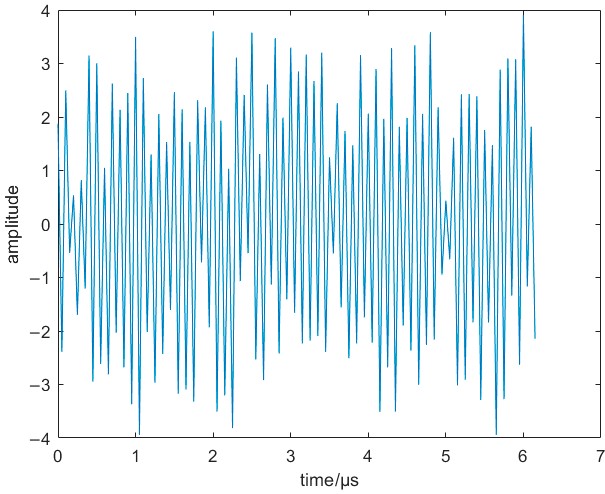

**Figure 3.** A continuous signal with noise emitted by an illuminator of opportunity.

The Doppler center and Doppler frequency rate are represented as $\overline{f}_{dc,k}$ and $\overline{f}_{dt,k}$, respectively. $\overset{\bullet}{R}_k(\eta)$ denotes the derivative of the bistatic range. Integrating this expression yields another formulation of the bistatic range:

$$R_k(\eta) = R_{0,k} - \frac{c}{f_c}\overline{f}_{dc,k}\eta - \frac{c}{2f_c}\overline{f}_{dt,k}\eta^2 \tag{4}$$

where $R_{0,k}$ represents the baseline distance at the reference time. Taking the range domain, Fourier transform of Equation (2) yields

$$S_{pc,P_k}(f_d, \eta) = \sigma_{\eta,k} R(f_d) \cdot \exp\left\{ -\mathrm{j}\frac{2\pi(f_d + f_c)}{c}\left[ R_{0,k} - \frac{c}{f_c}\overline{f}_{dc,k}\eta - \frac{c}{2f_c}\overline{f}_{dt,k}\eta^2 \right] \right\} \tag{5}$$

where $f_d$ represents the range-domain frequency, and $R(\cdot)$ represents the Fourier transform of $r(\cdot)$. This study establishes a coordinate system, as shown in Figure 1, where the position vector, velocity vector, and acceleration of the target at the reference time are denoted as $\mathbf{R}_T$, $\mathbf{v}_T$, and $\mathbf{a}_T$, respectively. The position vector, velocity vector, and acceleration of the k-th illuminator of opportunity are $\mathbf{R}_{P_k}$, $\mathbf{v}_{P_k}$, and $\mathbf{a}_{P_k}$, respectively. The velocity of the receiver parameter is denoted as $\mathbf{0}$ (in practice, it is represented as $\mathbf{v}_O$), and the actual motion parameters of the target and illuminators of opportunity need to be corrected based on the receiver's motion parameters. The bistatic range is represented as

$$R_k(\eta) = R_{O,T}(\eta) + R_{P_k,T}(\eta) - R_{O,k}(\eta) = \left| \mathbf{R}_T + \mathbf{v}_T\eta + \tfrac{1}{2}\mathbf{a}_T\eta^2 \right|$$
$$+ \left| \mathbf{R}_{P_k} + \mathbf{v}_{P_k}\eta + \tfrac{1}{2}\mathbf{a}_{P_k}\eta^2 - \mathbf{R}_T - \mathbf{v}_T\eta - \tfrac{1}{2}\mathbf{a}_T\eta^2 \right| \tag{6}$$
$$- \left| \mathbf{R}_{P_k} + \mathbf{v}_{P_k}\eta + \tfrac{1}{2}\mathbf{a}_{P_k}\eta^2 \right|$$

Taking the derivative of Equation (6) and comparing Equation (4) and Equation (6), the expression for the Doppler parameter of the echo at zero in the azimuth direction is

$$\overline{f}_{dc,k} = -\frac{f_c}{c}\frac{dR_k(\eta)}{d\eta}\bigg|_{\eta=0} = -\frac{f_c}{c}\left\{ |\mathbf{R}_T|^{-1}\cdot\mathbf{R}_T\mathbf{v}_T^{\mathrm{T}} + |\mathbf{R}_{P_k} - \mathbf{R}_T|^{-1}\cdot\left(\mathbf{R}_{P_k} - \mathbf{R}_T\right)\left(\mathbf{v}_{P_k} - \mathbf{v}_T\right)^{\mathrm{T}} - |\mathbf{R}_{P_k}|^{-1}\cdot\mathbf{R}_{P_k}\mathbf{v}_{P_k}^{\mathrm{T}} \right\} \tag{7}$$

$$\begin{aligned}
\overline{f}_{dt,k} &= -\frac{f_c}{c}\frac{d^2 R_k(\eta)}{d\eta^2}\bigg|_{\eta=0} = \\
&-\frac{f_c}{c}\Big\{ -|\mathbf{R}_T|^{-3}\cdot(\mathbf{R}_T\mathbf{v}_T)^2 + |\mathbf{R}_T|^{-1}(\mathbf{v}_T^2 + \mathbf{R}_T\mathbf{a}_T^{T}) \\
&- \left|\mathbf{R}_{P_k} - \mathbf{R}_T\right|^{-3}\left[(\mathbf{R}_{P_k} - \mathbf{R}_T)(\mathbf{v}_{P_k} - \mathbf{v}_T)^{T}\right]^2 \\
&+ \left|\mathbf{R}_{P_k} - \mathbf{R}_T\right|^{-1}\cdot\left[\left|\mathbf{v}_{P_k} - \mathbf{v}_T\right|^2 + (\mathbf{R}_{P_k} - \mathbf{R}_T)\left(\mathbf{a}_{P_k} - \mathbf{a}_T\right)^{T}\right] \\
&+ \left|\mathbf{R}_{P_k}\right|^{-3}\cdot(\mathbf{R}_{P_k}\mathbf{v}_{P_k}^{T})^2 - \left|\mathbf{R}_{P_k}\right|^{-1}\cdot\left(\left|\mathbf{v}_{P_k}\right|^2 + \mathbf{R}_{P_k}\mathbf{a}_{P_k}^{T}\right) \Big\}
\end{aligned} \tag{8}$$

When the baseline distance is relatively small, around 10 km, and the range between the target and the receiver is more than 100 km, both illuminators of opportunity and the target move along the corresponding position vector direction. The velocity of illuminators of opportunity is 10 m/s, while the target velocity is 100 m/s. The mathematical model simplifies this scenario as an isosceles triangle, as shown in Figure 4. The vertex angle of the isosceles triangle line is calculated using the cosine theorem, and the values of the carrier frequency and the velocity of light from Table 1 are used. By applying (8), the Doppler frequency rate can be calculated as 0.0424 Hz/s. The Doppler frequency rate does not change significantly even after altering the k value, indicating that the signal has approximately equal Doppler frequency rate in different environments. This assumption is supported by simulations, as shown in Figure 5.

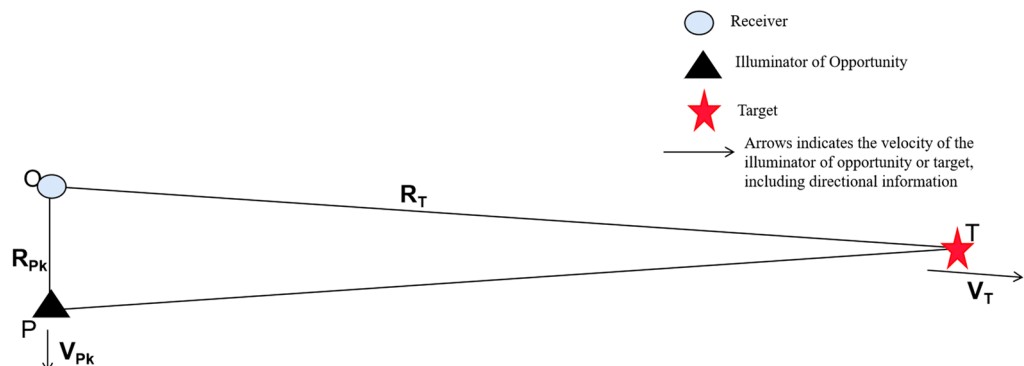

**Figure 4.** The mathematical model simplifies this scenario as an isosceles triangle.

**Table 1.** Other parameters.

| Name | Value |
|---|---|
| SNR/dB | −10 |
| Doppler frequency rate/(Hz/s) | −0.015 |
| Velocity of light/(m/s) | 299, 792, 458 |
| Carrier frequency/Hz | $1.26852 \times 10^9$ |
| Number of frames | N = 10 |
| Target observation time/s | 20 |
| Target complex scattering coefficient | 1 |

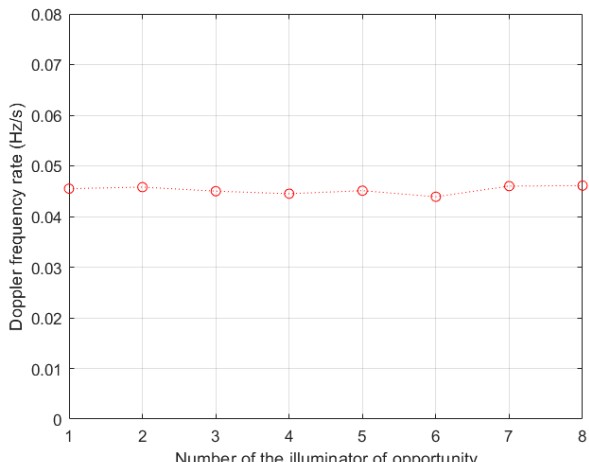

**Figure 5.** Doppler frequency rate varies with the different illuminators of opportunity.

According to the hypothesis, Equation (8) can be simplified to

$$\overline{f}_{dt} = \overline{f}_{dt,k} = -\frac{f_c}{c}\frac{d^2 R_k(\eta)}{d\eta^2}\bigg|_{\eta=0} \approx -\frac{f_c}{c}\left\{|\mathbf{R}_T|^{-1}\overline{v}_t^2 + \overline{a}_r\right\} \tag{9}$$

where $\overline{v}_t = |\mathbf{v}_T| \cdot \sin\alpha$ is the equivalent radial velocity, $\alpha$ is the angle between $\mathbf{R}_T$ and $\mathbf{v}_T$, $\overline{a}_r = |\mathbf{a}_T| \cdot \cos\vartheta$, and $\vartheta$ is the angle between $\mathbf{R}_T$ and $\mathbf{a}_T$.

## 3. Target Detection Method

Most studies currently focus on targets located in 2D planes [16,18,19] or static illuminators of opportunity [2,3]. This paper aims to detect targets in 3D space using a mobile PMR system model. The parameter space is expanded to include X-Y-Z-V [23], where X-Y-Z represents the target position parameter and V represents the target velocity parameter. The size of the parameter space ranges from M. In this parameter space, a series of cells

are arranged, assuming that the positions of different cells represent possible targets. A velocity group is assigned to each cell, enabling the determination of the possible target's position and velocity information using Equation (9). Subsequently, the bistatic range of the target can be obtained. By utilizing the bistatic range, the compressed signal is phase-compensated. Finally, through the accumulation of multiple illuminators of opportunity, the precise position of the actual target can be determined.

In the coordinate system shown in Figure 1, the motion target is located in the X-Y-Z space. The observation area of the X-Y-Z plane is divided into multiple grids, and the position of each cell is represented as (x,y,z). The range $|\mathbf{R}_T| = \sqrt{x^2 + y^2 + z^2}$ from the grid to the receiver, the observation angle $\theta = \arctan(\frac{\sqrt{x^2+y^2}}{z})$, and the projection angle $\varphi = \arctan(\frac{y}{x})$ onto the x-o-y plane can be obtained. Given the extracted Doppler frequency rate $\overline{f}_{dt}$ of the target, Equation (9) can be adopted to calculate the corresponding equivalent radial velocity for each grid cell.

$$v_t = \sqrt{-\overline{f}_{dt} \cdot \frac{c}{f_c} \cdot |\mathbf{R}_T| - |\mathbf{R}_T|\overline{a}_r{}^T} \tag{10}$$

Setting K velocities $v_r$, the velocity vector $v = (v_x, v_y, v_z)$ of possible targets in multiple grid cells in space can be calculated based on $v_t$, observation angles $\theta$ and $\varphi$, and acceleration $a_r$, where $sign[\cdot]$ stands for symbolic function. This paper considers the situation where the target is in high-velocity motion, with little maneuverability in a short period of time, and the acceleration can be ignored by setting the acceleration $\overline{a}_r = 0$.

$$\begin{cases} v_x = v_r \cos\theta \cos\varphi + sign[v_t]v_t \sin\theta \sin\varphi \\ v_y = v_r \cos\theta \sin\varphi + sign[v_t]v_t \sin\theta \cos\varphi \\ v_z = v_r \sin\theta + sign[v_t]v_t \cos\varphi \end{cases} \tag{11}$$

At this stage, the position and velocity of each cell in the space are determined, which serve as the potential target. Subsequently, the position and velocity information is incorporated into Equation (6) to calculate the bistatic range $\hat{R}_k(\eta; x, y, z, v)$ for different illuminators of opportunity. The bistatic range is then incorporated into Equation (2) to derive the signal formula that is associated with azimuth time, target position, and velocity.

$$s_{P_k}(\eta; x, y, z, v) = \sigma_{\eta,k} \exp\left\{-j2\pi f_c \frac{\hat{R}_k(\eta; x, y, z, v)}{c}\right\} \times r\left[\frac{\hat{R}_k(\eta; x, y, z, v)}{c}\right] \tag{12}$$

In order to account for the variation of the complex scattering coefficient caused by target motion, the azimuth signal is divided into N frames. Within each frame, denoted as $\hat{s}_{P_{k,n}}(\eta; x, y, z, v)$, the complex scattering coefficient can be considered constant over a very short time interval. Phase compensation is then applied based on the bistatic range information in the received echo, enabling for azimuth time coherent accumulation.

$$s_{P_k,n}(x, y, z, v) = \sum_{\eta} \hat{s}_{P_k,n}(\eta; x, y, z, v) \exp(\frac{j2\pi R_k(\eta)}{\lambda}) \tag{13}$$

where $R_k(\eta)$ represents the bistatic range information in the range pulse compression signal. Afterward, non-coherent summation is performed on the inter-frame signals to obtain

$$s_{P_k}(x, y, z, v) = \sum_{n} |s_{P_k,n}(x, y, z, v)| \tag{14}$$

The bistatic range information of all cells in the parameter space is searched and matched. The results are then projected onto the parameter space. The incoherent accu-

mulation and summation of multiple signals of illuminators of opportunity in the spatial parameter domain yield the following results

$$s(x, y, z, v) = \sum_k \left| s_{P_k}(x, y, z, v) \right| \tag{15}$$

A demonstration simulation of the above operations is as follows. At a SNR condition of 10 dB, three illuminators of opportunity independently detect the target. Despite having different observation configurations, it is determined that the three targets are actually the same target. After extracting the Doppler frequency rate of the target, a space search is conducted, and the phase of the echo is compensated at the corresponding position. As a result, the target is aligned, as depicted in Figure 6, and the SNR is enhanced after the alignment fusion process [24].

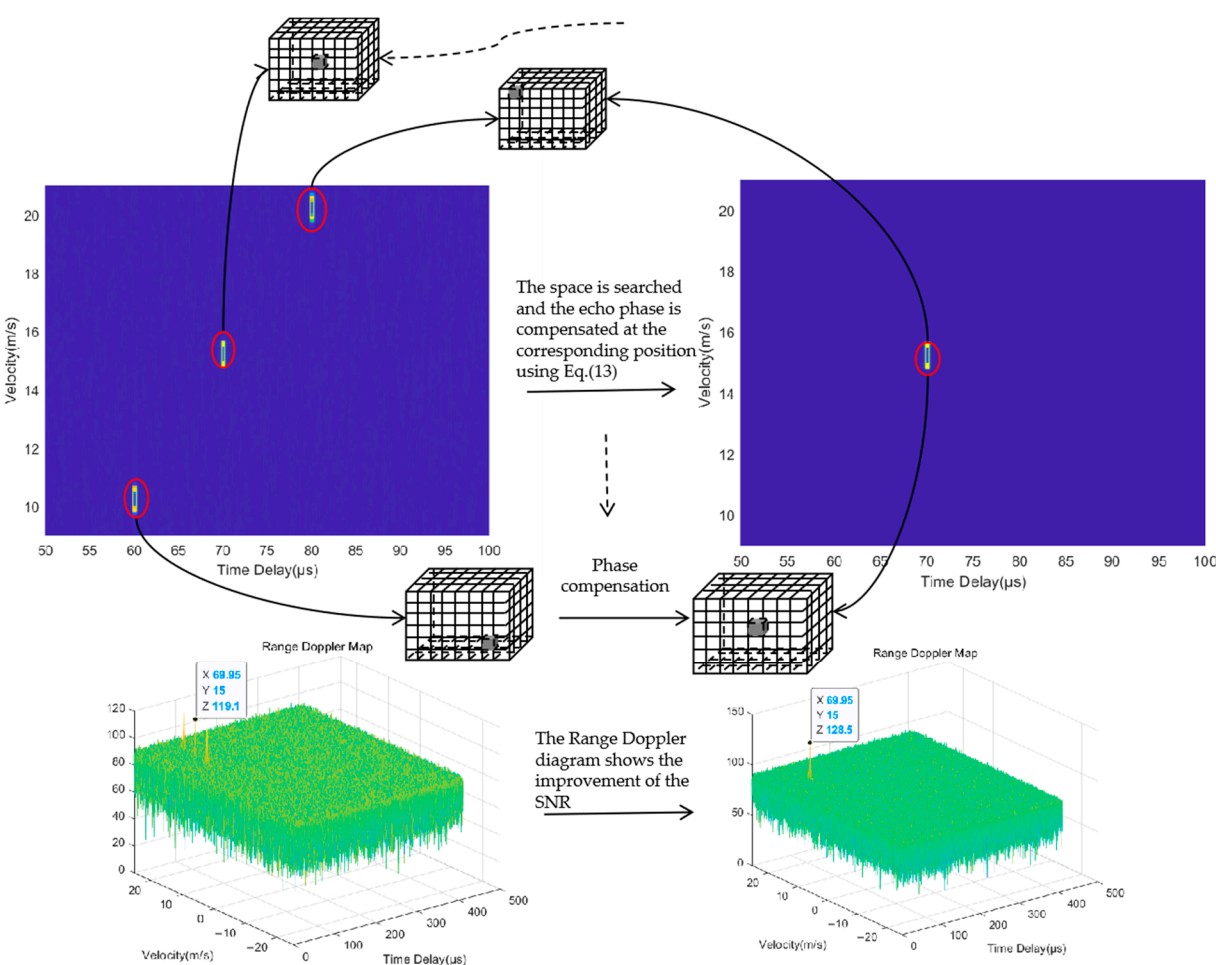

**Figure 6.** Schematic drawing about aligning and fusion.

The position and velocity of the target are determined based on the peak position in the X-Y-Z-V multiple parameters spatial overlay result, where the X-Y-Z spatial coordinates are represented as a searchable one-dimensional array, and all spatial grid cell information is stored as shown in (16). If the final target detection position is 111, the corresponding parameter space position is (2, 2, 1). The algorithm flow chart of target detection method is shown in Figure 7.

$$(x, y, z) : 100(x - 1) + 10(y - 1) + z \tag{16}$$

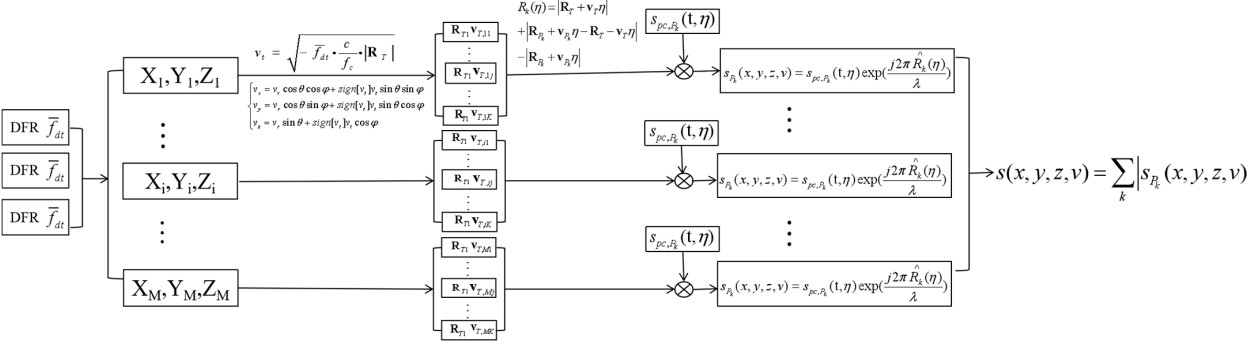

**Figure 7.** The algorithm flow chart of target detection method. Three illuminators of opportunity extract the same Doppler frequency rate. M represents the number of cells in the space, and K represents a set of velocity numbers. This is conducive to calculating the complexity of the subsequent algorithm.

The pseudocode of the target detection method based on multi-parameter space for mobile PMR is presented in Algorithm 1.

---

**Algorithm 1**: The target detection method based on multi-parameter space for mobile PMR

---

**Require:**

A range pulse compression signal from the direct signal and the echo signal: $S_{pc,P_k}(t,\eta)$

The Doppler frequency rate is extracted from a range pulse compression signal: $\bar{f}_{dt}$

1: **for** i = 1:1:M, j = 1:1:K, n = 1:1:N **do**

2:    The range $\mathbf{R}_T$ from the cell to the receiver is calculated by $|\mathbf{R}_T| = \sqrt{x^2 + y^2 + z^2}$.

3:    The equivalent radial velocity $v_T$ is calculated by (10).

4:    The velocity in the current cell is calculated by using the specified value for the j group velocity and the (11).

5:    The bistatic range is calculated according to the obtained range $\mathbf{R}_T$ and velocity $\mathbf{v}_T$ and (6).

6:  The phase compensation is performed using (13), and n-frame signals are simultaneously accumulated.

7:  The summation of multiple signals of illuminators of opportunity in the spatial parameter domain is calculated according to (15).

8: **end for**

9: The peak position is the location of the target.

10: **return** Position parameter (x,y,z)

---

## 4. Simulation Experiment

The simulation scenario is configured with the illuminators of opportunity and receiver located on a mobile platform, while the target is in 3D space and at a range of over 100 km away from both the illuminators of opportunity and receiver. Table 2 presents the positions, velocities, accelerations of the target and illuminators of opportunity. Considering the limitation of a double base angle, the study focuses on far-field targets located at a range of over 100 km. It is assumed that the target has zero acceleration and a grid cell range of $10 \times 10 \times 10$. Other parameters are shown in Table 1. The bistatic range, calculated using Equation (6), is contained in the range pulse compression signal information, which can be used for phase compensation during subsequent spatial grid cell searches. According to Formula (9), the Doppler frequency rate of the target is independent of the k value, meaning that Doppler frequency rate does not change with the change of observation configuration. The Doppler frequency rate of the target is extracted as $-0.015$ Hz/s, as shown in Figure 8. The equivalent radial velocity of the target is calculated using Equation (9), and the range and velocity corresponding to each grid cell are computed, along with the dual baseline distances that need to be traversed.

The results of the traversal illuminators of opportunity are projected onto the X-Y-Z-V space individually, as shown in Figures 9–11. The target is submerged in noise, and each figure shows multiple possible target results from the traversal. The incoherent

accumulation is performed on the results of three illuminators of opportunity, as depicted in Figure 12. According to the final accumulated result, the target signal accumulates in the X-Y-Z-V space, resulting in a single peak at coordinates (223, 120, 1). As per (16), the position information for 223 is queried. The X-Y-Z space coordinates of 223 are (3, 3, 3), which correspond to the target coordinates of (60, 80, 100) km. Additionally, the radial velocity is measured to be 120 m/s. The target velocity is calculated to be (−38.2, −58.3, −71.3) m/s using Equation (11).

**Table 2.** Various simulation parameters.

| Name | Position (km) | Velocity (m/s) | Acceleration (m/s$^2$) |
|---|---|---|---|
| Illuminator of opportunity 1 | (20, 19, 0) | (20, 10, 0) | (0.1, 0.1, 0) |
| Illuminator of opportunity 2 | (10, 15, 0) | (10, 20, 0) | (0.1, 0.1, 0) |
| Illuminator of opportunity 3 | (5, 10, 0) | (10, 10, 0) | (0.1, 0.1, 0) |
| Target | (60, 80, 100) | (−40, −60, −70) | (0, 0, 0) |

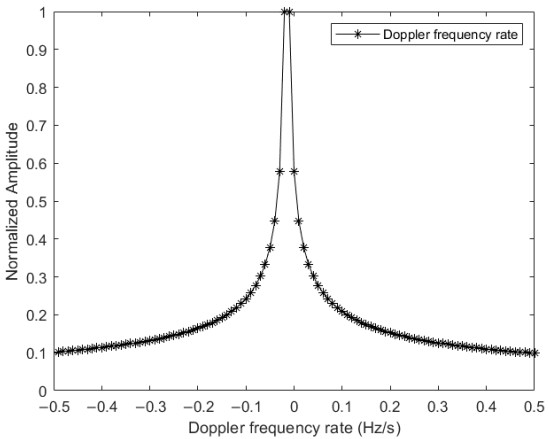

**Figure 8.** Extracting the Doppler frequency rate of a target.

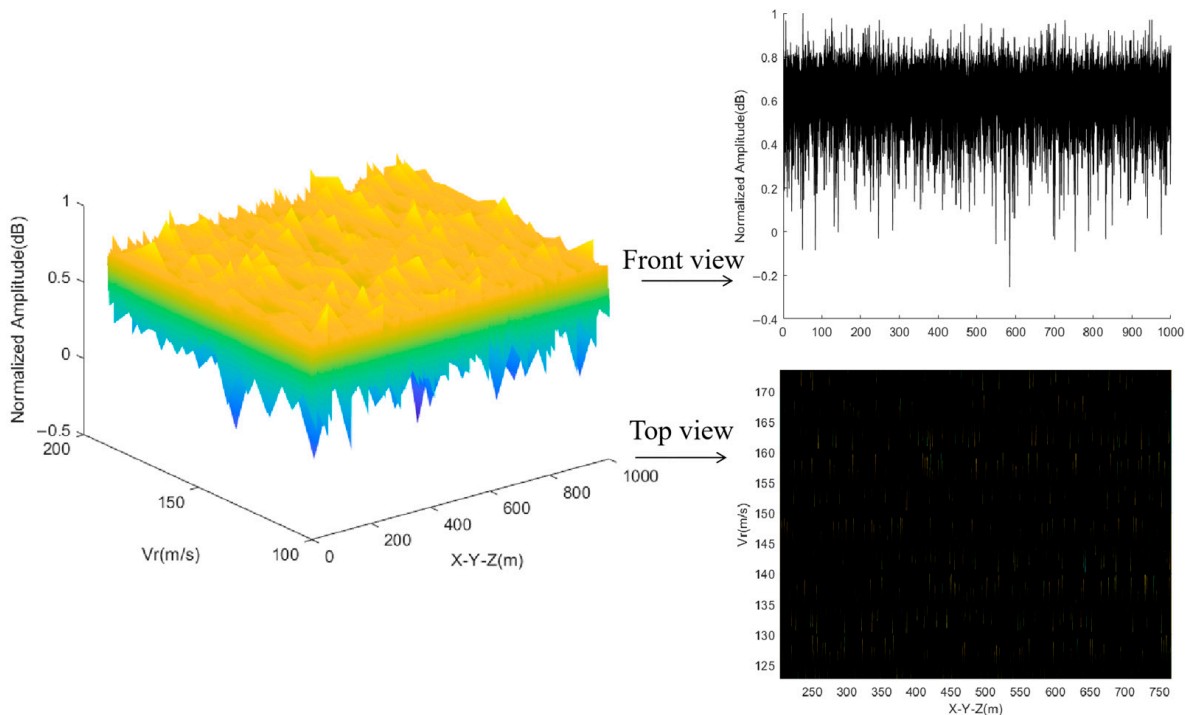

**Figure 9.** Accumulated result from illuminator of opportunity 1.

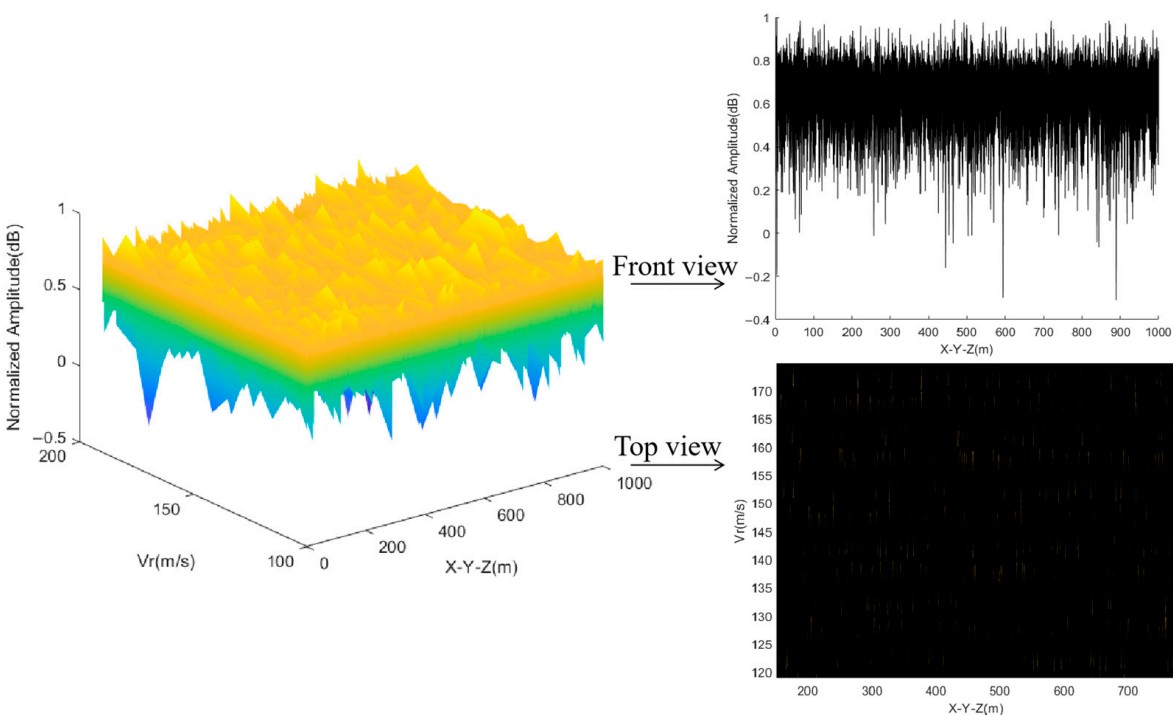

**Figure 10.** Accumulated result from illuminator of opportunity 2.

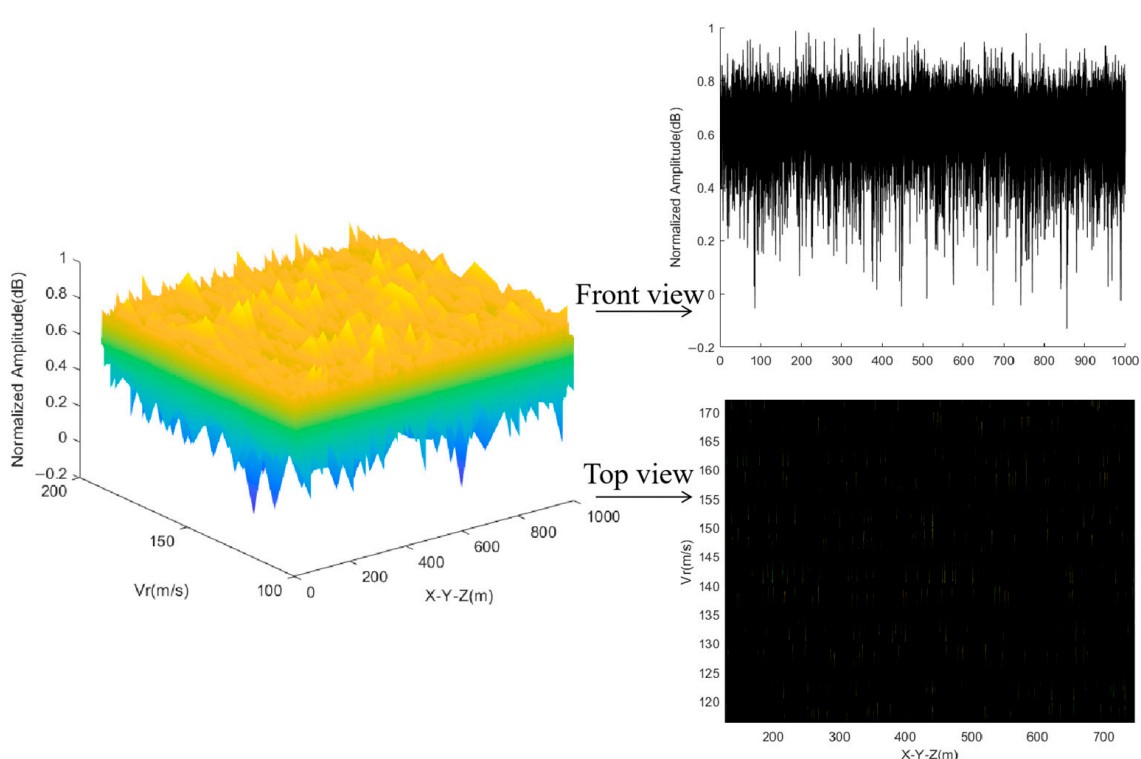

**Figure 11.** Accumulated result from illuminator of opportunity 3.

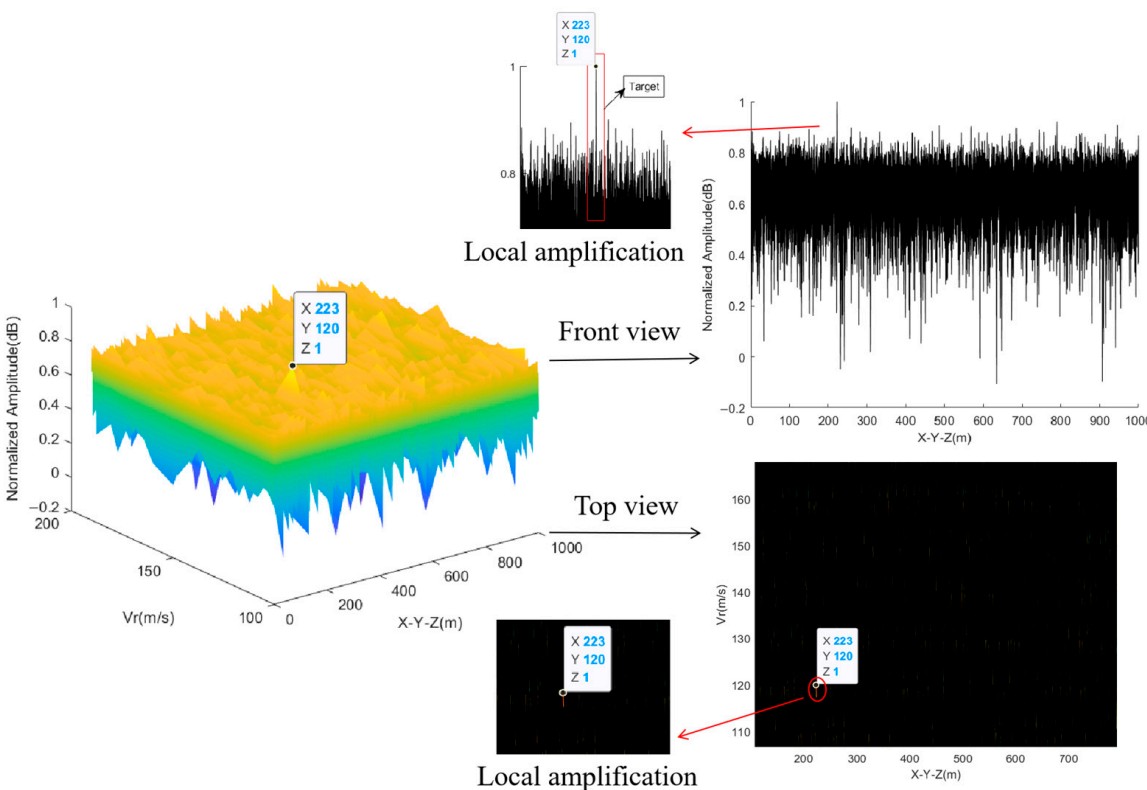

**Figure 12.** The fused results of multiple illuminators of opportunity.

As shown in Figures 9–11, when each illuminator of opportunity detects the target separately, the target located at 233 is submerged in noise, or a ghost target appears. The fusion results of multiple illuminators of opportunity in Figure 12 demonstrate an improvement in the SNR. The SNR is calculated as follows

$$SNR = 10\log\left(\frac{P_s}{P_n}\right) \tag{17}$$

where $P_s$ represents signal power and $P_n$ represents noise power. According to Formula (17), the SNR of a single illuminator of opportunity detection target is 9.43 dB, while the SNR of multiple illuminators of opportunity fusion is 11.14 dB, resulting in an increase of 18%. The threshold is set to 0.9, which is higher than this value can be judged as a target. It can be observed that a single illuminator of opportunity has false detection when detecting targets, and there are numerous ghost targets. However, the accuracy and precision of multiple illuminators of opportunity detection show a significant increase. This comparison is illustrated in Figure 13. This demonstrates a significant enhancement in detection probability and SNR through the fusion method.

Comparing the result with the multiple illuminators of opportunity fusion 2D sea surface target detection in [25], the target position parameter was set to (1000, 0, 0) m, and the velocity parameter was set to (7.014, 7.014, 0) kn. As shown in Figures 14 and 15, the target detection position is (1002, −8, 0) m and the target velocity is (7.06, 6.96, 0) kn [25]. Error calculation is shown in Equation (18), where x represents the exact value and x* represents the approximation.

$$error = \frac{x - x*}{x} \tag{18}$$

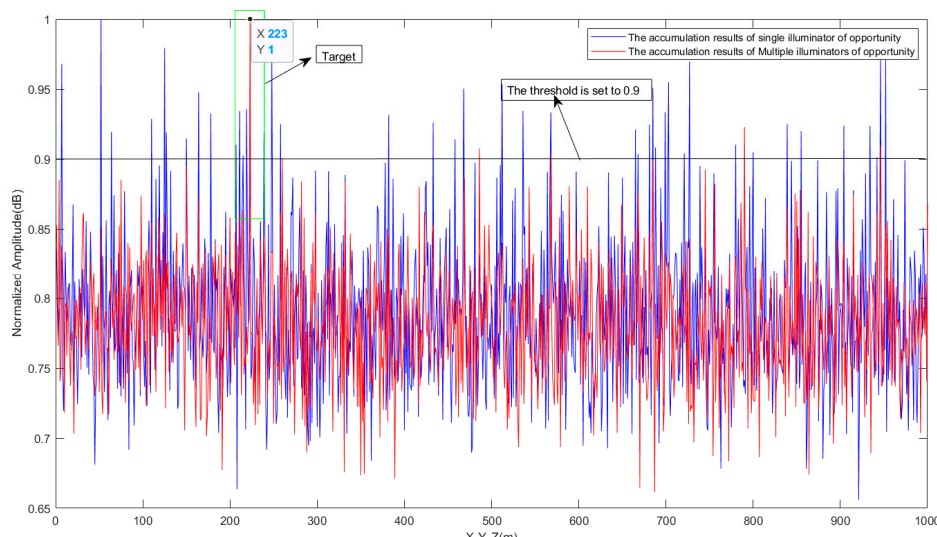

**Figure 13.** Comparison of single illuminator of opportunity and multiple illuminators of opportunity fusion results.

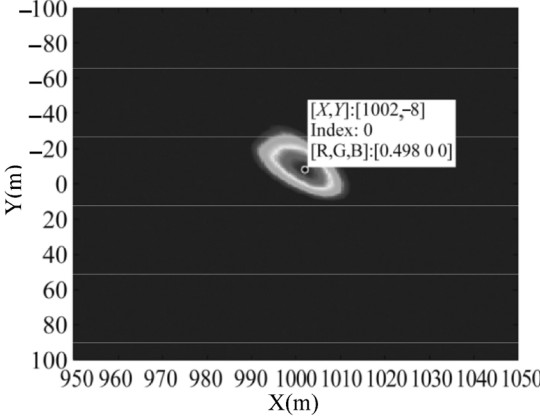

**Figure 14.** The position detection results of 2D sea surface target.

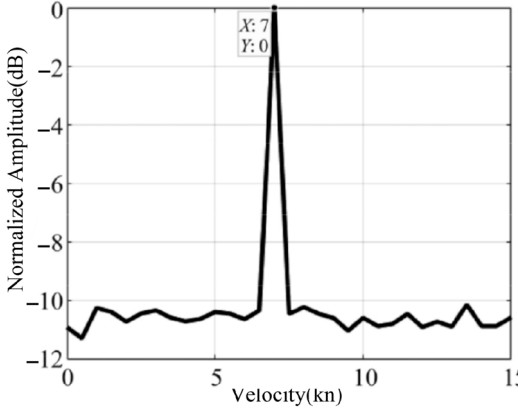

**Figure 15.** The velocity estimation results of the 2D sea surface target.

The detection target error was 1%, and the velocity measurement error was 0.7%. In this paper, the 2D space target scene is transformed into a 3D space target scene. The proposed method accurately detects the target position, and the velocity measurement error is 2.8%. The Z-axis component dimension of the velocity increases in three-dimensional space. This component is also influenced by the direction angle, and any measurement error

in the direction angle will further contribute to the error in the target velocity. Considering the significant range between the target and the receiver, as well as the close proximity between the receiver and the illuminators of opportunity in the scenario, the dual baseline angle is relatively small. This implies that the approximation of the Doppler frequency rate is more justifiable. The equivalent radial velocity extracted is more accurate, leading to smaller errors in the position detection and velocity measurement results.

As shown in Figure 16, the comparison of the proposed method, the method proposed in [25] which is 2D sea surface target detection method and the non-fusion method shows that the output SNR of the proposed method is significantly improved. Under the condition of −10 dB, the output SNR of the proposed method is 1.71 dB higher than that of the 2D sea surface target detection method and 3.84 dB higher than that of the non-fusion method, which is consistent with the actual fusion increase in the three illuminators of opportunity. Figure 17 shows the detection probabilities of the proposed method compared with the 2D sea surface target detection method and non-fusion method. For the same detection probability (90%), the proposed method, the 2D sea surface target detection method, and the non-fusion method require SNRS of −2.382 dB, 0.333 dB, and 1.753 dB, respectively, as shown in Table 3. However, it is evident that as the number of dimensions increases, the number of parameters to be estimated also increases proportionally, resulting in an increased calculation burden for the algorithm.

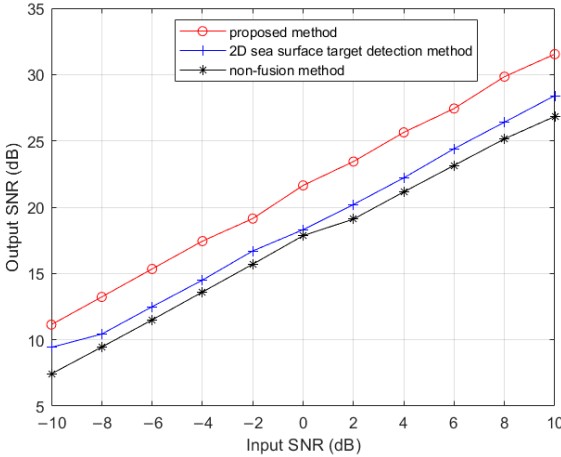

**Figure 16.** Comparison of proposed method, 2D sea surface target detection method and non-fusion method.

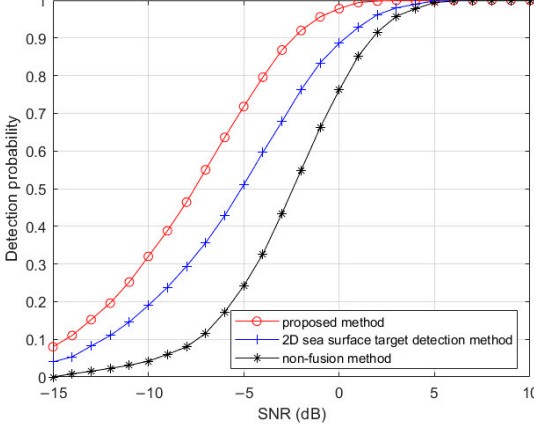

**Figure 17.** The detection probability of the proposed method compared with the 2D sea surface target detection method and non-fusion method.

**Table 3.** The SNRs required for the same detection probability (90%).

| Name | SNR (dB) |
|---|---|
| Proposed method | −2.382 |
| 2D sea surface target detection method | 0.333 |
| Non-fusion method | 1.753 |

Regarding the computational complexity, the detection target in this paper is extended from 2D to 3D space, and the target parameter is changed from X-Y-V to X-Y-Z-V, which leads to an approximately twofold increase in the computational complexity and a significant rise in the computation time. However, this trade-off is acceptable because the proposed method in this paper has remarkably enhanced the SNR. The computational complexity of each method is shown in Table 4. Figure 18 more intuitively compares the computational complexity of the three methods when M is different. In the subsequent work, a two-step method can be employed to shrink the search area by minimizing the power consumption, followed by a search algorithm, which can reduce the computational complexity.

**Table 4.** The computational complexity of each method.

| Name | Computational Complexity |
|---|---|
| Proposed method | $O(n^{M+K+N})$ |
| 2D sea surface target detection method | $O(n^{\frac{2}{3}M+K+N})$ |
| Non-fusion method | $O(n^{\frac{2}{3}M+K})$ |

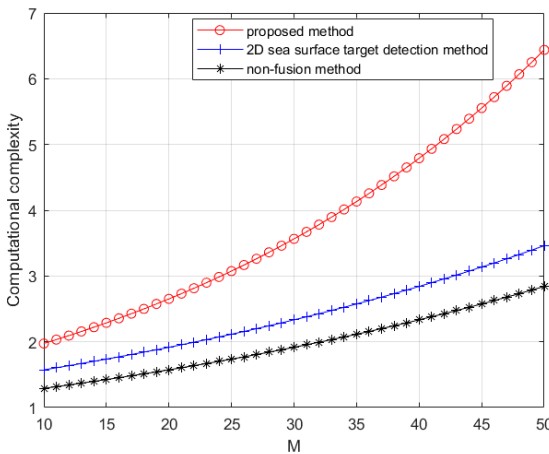

**Figure 18.** The computational complexity of the three methods.

## 5. Discussion

Several potential limitations or challenges in our work need to be indicated. Firstly, the maneuverability of the high-velocity target is limited, but this does not have a great impact. It is only necessary to increase the radial acceleration dimension, but the calculation amount will be increased again. Secondly, when obtaining the Doppler frequency rate information of the target, it is assumed that the double base angle is very small, the distance of the illuminators of opportunity is very close, and the target is far away. The near-field target cannot be detected by the proposed method. Thirdly, when searching parameter space, the division of space units determines the calculated quantity and resolution. Too many cells will increase the calculation amount, and too few cells will reduce the resolution of the multi-object. The number of cells should be determined according to the situation. Furthermore, it is important to note that this paper does not consider the coverage of illuminators of opportunity. The geometry of the PMR can also influence the detection

target [26]. Lastly, in the case of multiple targets, the Doppler frequency rate values of different targets are very close to each other. The Doppler frequency rate extracted in this paper is $-0.015$ Hz/s, which results from the target being far away and the bistatic angle being small. Thus, it is challenging to distinguish multiple targets, which is also the future research direction.

The scenario in this paper is aimed at a moving network radar. When part of the active radar in the network radar fails, this system can be used to improve the detection probability or to consider civil marine radars (CMR) on merchant ships as illuminators of opportunity in a PMR configuration [24]. The proposed method provides a new solution for the fusion detection of targets by PMR on 3D mobile platforms. It has great potential application value in an active radar network system.

## 6. Conclusions

This paper proposes a novel target detection method based on multi-parameter space for mobile PMR. The proposed method can detect mobile 3D spatial targets with multiple illuminators of opportunity and receivers. By introducing a multi-parameter spatial method and using the extracted Doppler frequency rate of the target to compensate phase, this paper performs to search the amplitude maximum through the multi-parameter spatial method. Meanwhile, the accuracy of localization and SNR are improved through multiple steps such as azimuth-time accumulation, frame segmentation, and data fusion. Additionally, the detection of a moving single target is simulated. Compared to a single illuminator of opportunity, the SNR of three-signal detection is enhanced by 1.71 dB, showing significant improvement through incoherent fusion. The mobile PMR system model proposed in this paper exploits the target's Doppler frequency rate information to achieve 3D spatial target detection and velocity estimation. This provides a new solution for the fusion detection of targets by PMR on 3D mobile platforms.

**Author Contributions:** Conceptualization, H.Z., X.W., X.L., Q.D. and Y.L.; Investigation, H.Z., X.W. and Q.D.; Software, H.Z.; Writing—original draft preparation, H.Z. and X.W.; Writing—review and editing, H.Z., X.W., X.L. and Y.L.; Data curation, N.L., K.C.; Resources, N.L., K.C.; Project administration, X.L.; Validation, H.Z., X.L., Q.D., N.L., K.C., X.W. and Y.L. All authors have read and agreed to the published version of the manuscript.

**Funding:** This research was supported by the Basic Research Foundation.

**Data Availability Statement:** Data sharing not applicable.

**Acknowledgments:** The authors would like to thank the staff of the National Key Laboratory of Microwave Imaging Technology, Aerospace Information Research Institute, Chinese Academy of Sciences, for their valuable conversations and comments.

**Conflicts of Interest:** The authors declare no conflict of interest.

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
