# Peer review of "A Novel Target Detection Method Based on Multi-Parameter Space for Mobile Passive Multistatic Radar"

_remotesensing, doi:10.3390/rs15204961_

Round 1
Reviewer 1 Report
In this paper, a 3D mobile PMR model is established and a detection method that enhances the SNR by fusing the signals from multiple IOO is proposed. Overall the paper is significant to be published, while some detail comments to improve the quality of paper is given as follows.
1. It may be hard to read if too much uncommon used acronyms are employed, e.g. IOO, DC, DFR etc. These kinds of acronyms can be written in full expression to some extent.
2. In the Line 34, the statement is advised to cite some literatures, e.g.
[R1] Yavari, Ehsan. Distortion reduction and signal estimation in Doppler radar physiological monitoring systems[D].University of Hawaii at Manoa, 2015.
[R2] Jinxin Sui; Jun Wang; Luo Zuo; Jie Gao. Cascaded Least Square Algorithm for Strong Clutter Removal in Airborne Passive Radar[J].IEEE Transactions on Aerospace and Electronic Systems,2022,Vol.58(1): 679-696.
On the other hand, besides the Doppler distortion and coupling mentioned in the paper, the Doppler resolution also seriously influences the estimation of Doppler, which is suggested to refer to the following works:
[R3] Jiahuan Wang; Pingzhi Fan; Yang Yang; Yongliang Guan. Range/Doppler Sidelobe Suppression in Moving Target Detection Based on Time-Frequency Binomial Design[C].2019 IEEE 30th Annual International Symposium on Personal, Indoor and Mobile Radio Communications (PIMRC),2019, 1-5.
[R4] Jiahua Zhu; Yongping Song; Nan Jiang; Zhuang Xie; Chongyi Fan; Xiaotao Huang. Enhanced Doppler Resolution and Sidelobe Suppression Performance for Golay Complementary Waveforms[J]. Remote Sensing, 2023, 15: 2452.
3. Please list the main contributions of the paper in a separate paragraph.
4. How to calculate the value of DFR in Line 156 without the information of fc?
5. What is the meaning of IOO’ in the paper?
6. The computing burden is suggested to be illustrated in real time besides expressed in O(·), which would be more intuitively for comparison.
7. Please unify the font type and size of all the equations, and it is not necessary to plot such big size of the figures.
Reviewer 2 Report
This paper proposes a spatial target detection method for mobile multi-static passive radars. It is a hot topic in the community of passive radars. The paper is well-written, well-organized, and balanced in its structure. The algorithm proposed seems valid and the results obtained prove its effectiveness. The proposed methodology is innovative and could be accepted, but it will require some minor revisions.
1) Is the modeling method proposed suitable for application to the scenario with multi-target and maneuvering target detection? it is recommended to elaborate on this aspect in the paper.
2) Compared to the literature [11]、[21], the improvement of the 3D spatial modeling proposed in this paper needs to be highlighted and the differences with the methodology in the literature need to be clarified.
3) The Accumulated result of IOO in Figures 9-12 is not distinct, and it is recommended to replace it with a clear one or to label it.
4) The conclusion needs to state clearly whether the goals (defined in the introduction) were achieved, and the reasons. Some outlook and future work can also be mentioned (for instance, the limitations that the method proposed in this paper should be addressed).
5) It is recommended that this paper add a description of the simulation configuration and data type (for example, the data processed in this paper is the range-doppler domain of the signal).
Reviewer 3 Report
The research proposed the novel detection method with lidar. Overall, the research makes sense. However the writting must be improved.
(1) Introduction should be more focused on review state of art detection method
(2) You do not need put graphs and mathematics models in introduction
(3) Methods includes a flow diagram to show the entire flow.
(4) Please share your simulation code. I want to see whether the simulation makes sense
(5) Some Figures in results section must be improved such as Figure among others.
Please find way to improve english.
Round 2
Reviewer 1 Report
The authors have addressed all my comments in the revision, I think it is worth to be published.